# Small Cell Carcinoma of the Ovary, Hypercalcemic Type (SCCOHT) beyond *SMARCA4* Mutations: A Comprehensive Genomic Analysis

**DOI:** 10.3390/cells9061496

**Published:** 2020-06-19

**Authors:** Aurélie Auguste, Félix Blanc-Durand, Marc Deloger, Audrey Le Formal, Rohan Bareja, David C. Wilkes, Catherine Richon, Béatrice Brunn, Olivier Caron, Mojgan Devouassoux-Shisheboran, Sébastien Gouy, Philippe Morice, Enrica Bentivegna, Andrea Sboner, Olivier Elemento, Mark A. Rubin, Patricia Pautier, Catherine Genestie, Joanna Cyrta, Alexandra Leary

**Affiliations:** 1Medical Oncologist, Gynecology Unit, Lead Translational Research Team, INSERM U981, Gustave Roussy, 94805 Villejuif, France; Aurelie.auguste@gustaveroussy.fr (A.A.); Audrey.leformal@gustaveroussy.fr (A.L.F.); 2Gynecological Unit, Department of Medicine, Gustave Roussy, 94805 Villejuif, France; felix.blancdurand@gmail.com (F.B.-D.); beatrice.brunn@gustaveroussy.fr (B.B.); sebastien.gouy@gustaveroussy.fr (S.G.); philippe.morice@gustaveroussy.fr (P.M.); enrica.bentivegna@gustaveroussy.fr (E.B.); patricia.pautier@gustaveroussy.fr (P.P.); 3Bioinformatics Core Facility, Gustave Roussy Cancer Center, UMS CNRS 3655/INSERM 23 AMMICA, 94805 Villejuif, France; marc.deloger@gustaveroussy.fr; 4Caryl and Israel Englander Institute for Precision Medicine, Weill Cornell Medicine, New York, NY 10001, USA; rob2056@med.cornell.edu (R.B.); dcw2001@med.cornell.edu (D.C.W.); ans2077@med.cornell.edu (A.S.); ole2001@med.cornell.edu (O.E.); joanna.cyrta@gmail.com (J.C.); 5Institute for Computational Biomedicine, Weill Cornell Medicine, New York, NY 10001, USA; 6Genomic Platform Gustave Roussy Cancer Institute, 94805 Villejuif, France; Catherine.RICHON@gustaveroussy.fr (C.R.); olivier.caron@gustaveroussy.fr (O.C.); 7Department of Pathology, Hospital de la Croix Rousse, 69000 Lyon, France; mojgan.devouassoux@chu-lyon.fr; 8Department of Pathology and Laboratory Medicine, Weill Cornell Medicine, New York, NY 10001, USA; 9Department for BioMedical Research, University of Bern, 3001 Bern, Switzerland; mark.rubin@dbmr.unibe.ch; 10Department of Pathology, Gustave Roussy, 94805 Villejuif, France; catherine.genestie@gustaveroussy.fr; 11Department of Pathology, Institut Curie, Universite Paris Sciences et Lettres, 6 rue d’Ulm, 75005 Paris, France

**Keywords:** ovary, small cell carcinoma, hypercalcemic, SMARCA4, SWI/SNF

## Abstract

Small cell carcinoma of the ovary, hypercalcemic type (SCCOHT) is an aggressive malignancy that occurs in young women, is characterized by recurrent loss-of-function mutations in the *SMARCA4* gene, and for which effective treatments options are lacking. The aim of this study was to broaden the knowledge on this rare malignancy by reporting a comprehensive molecular analysis of an independent cohort of SCCOHT cases. We conducted Whole Exome Sequencing in six SCCOHT, and RNA-sequencing and array comparative genomic hybridization in eight SCCOHT. Additional immunohistochemical, Sanger sequencing and functional data are also provided. SCCOHTs showed remarkable genomic stability, with diploid profiles and low mutation load (mean, 5.43 mutations/Mb), including in the three chemotherapy-exposed tumors. All but one SCCOHT cases exhibited 19p13.2-3 copy-neutral LOH. *SMARCA4* deleterious mutations were recurrent and accompanied by loss of expression of the *SMARCA2* paralog. Variants in a few other genes located in 19p13.2-3 (e.g., *PLK5*) were detected. Putative therapeutic targets, including *MAGEA4*, *AURKB* and *CLDN6*, were found to be overexpressed in SCCOHT by RNA-seq as compared to benign ovarian tissue. Lastly, we provide additional evidence for sensitivity of SCCOHT to HDAC, DNMT and EZH2 inhibitors. Despite their aggressive clinical course, SCCOHT show remarkable inter-tumor homogeneity and display genomic stability, low mutation burden and few somatic copy number alterations. These findings and preliminary functional data support further exploration of epigenetic therapies in this lethal disease.

## 1. Introduction

Small cell carcinoma of the ovary of the hypercalcemic type (SCCOHT) is a rare, highly aggressive tumor that affects mainly young women (median age: 24 years). Prognosis is poor, as most patients die within two years of diagnosis [1]. The histogenesis of SCCOHT remains unclear, although there is increasing evidence in favor of a germ cell origin [2,3]. In addition, it has been proposed that SCCOHT may represent the ovarian counterpart of malignant rhabdoid tumors [4].

While there is no international consensus regarding the optimal treatment of SCCOHT, it usually involves multimodal chemotherapy, radical surgery and possibly, radiotherapy [5]. However, no randomized studies have been conducted to date and the available data consist of case reports or small retrospective series with heterogeneous management strategies. The only prospective clinical study in SCCOHT, a multicenter phase II trial conducted at Institut Gustave Roussy, tested combination chemotherapy (PAVEP: cisplatin, adriamycin, vepeside and cyclophosphamide) followed by radical surgery and high dose chemotherapy with autologous stem cell transplant, and demonstrated a three year survival rate of 49% among 27 SCCOHT patients [6]. This shows that even with intensive regimens, prognosis remains dismal, and that despite frequent initial response to chemotherapy, relapses are almost inevitable and tend to be refractory to second line chemotherapy.

The literature describing the genomic features of SCCOHT was scarce until 2014, when four groups identified loss-of-function mutations in *SMARCA4* (Brahma-related gene 1, BRG1) as a highly recurrent event in SCCOHT [3,4,7,8]. *SMARCA4* encodes one of the two possible catalytic subunits of the Switch/Sucrose Non-Fermentable (SWI/SNF) chromatin-remodeling complex. Others have since confirmed this finding, with *SMARCA4* mutations being found in over 90% of cases [9]. Nevertheless, effective treatment options to target this rare and lethal disease are still lacking.

We aimed to conduct an integrated genomic analysis of an independent cohort of SCCOHT by WES, RNA-Seq and aCGH to check for the presence of additional recurrent genomic alterations, which could allow the proposal of alternative treatment strategies.

## 2. Materials and Methods

### 2.1. Patients and Samples

Fresh-frozen tumor samples from 8 patients with SCCOHT were identified from the tumor banks of Institut Gustave Roussy, Cochin University Hospital, Grenoble University Hospital, Longjumeau University Hospital and Hôpital de la Croix Rousse. Central review for histological diagnosis was conducted by an expert pathologist. Matched blood samples were available for 6 patients. All patients provided written informed consent allowing the use of their tumor and non-tumor tissues for research. Approval from the hospital’s institutional review board was obtained for the study and funding was obtained via an educational grant awarded by the Foundation Gustave Roussy (local IRB approval RT12014). In addition, a further 33 formalin-fixed paraffin-embedded (FFPE) SCCOHT samples were available for validation studies. Tumors were obtained with patient consent and all data were anonymized.

### 2.2. DNA Extraction

DNA was extracted from fresh-frozen tumors and matched blood using the AllPrep DNA Mini Kit (Qiagen, Valencia, CA, USA) according to the manufacturer’s instructions, and quantified using Qubit (Thermo Fisher Scientific, Waltham, MA, USA). DNA integrity was measured using an Agilent BioAnalyzer (Agilent, Santa Clara, CA, USA).

### 2.3. Whole Exome Sequencing (WES)

Exome capture and library preparation were performed using the Sure Select Human All Exome v5 and SureSelectXT kits, respectively (Illumina, Agilent Technologies, CA, USA). Sequencing was done on matched tumor and normal samples using HiSeq2000 (Illumina, San Diego, CA, USA) in paired-end mode with a mean target depth of 100X. Reads were mapped using BWA-MEM (V0.7.5a-r405) [10] against reference genome hg19. Analysis of coverage was done using GATK (2.7.4-g6f46d11) [11] Depth of Coverage. Local realignment was performed around indels using GATK Realigner Target Creator and GATK Indel Realigner.

Variants were called with Varscan 2 [12], using hg19 as the reference genome and requiring a minimum tumor read depth of 6, a minimum somatic read depth of 8 and a minimum tumor allelic frequency of 0.10. Results were then annotated using SnpEff (4.3t) [13] and SnpSift (4.3t) [14] with dbSNP (v150_hg19) (http://www.ncbi.nlm.nih.gov/SNP/) and dbNSFP (v2.9) [15].

Tumor mutation burden (TMB) was calculated based on the number of non-synonymous, somatic-only mutations (single nucleotide variants and small insertions/deletions) with a somatic p-value threshold at <0.05 per megabase in coding regions considered as having sufficient coverage (6× in tumors and 8× in matched normal samples) by the variant caller [16].

Specific germline mutation analysis could not be performed, because a clause pertaining to germline testing was not included in the consent form at the time when each patient’s consent was obtained.

### 2.4. Oligonucleotide CGH Microarrays

DNA was labeled and hybridized, and CGH microarray analysis was performed as detailed in Appendix A. Resulting log2 (ratio) values were segmented using the CBS [17] algorithm implementation from the DNA copy package for R. Aberration status calling was automatically performed for each profile according to its internal noise (absolute variation of log2 (ratio) values across consecutive probes on the genome). All genomic coordinates were established on the UCSC Homo sapiens genome build hg19 [18].

### 2.5. SMARCA2 Promoter Sequencing

Sanger sequencing of *SMARCA2* promoter polymorphism sites was performed on DNA from 8 fresh-frozen SCCOHT tumor samples and 2 cell lines (BIN67 and SCCOHT-1). The following primers were used: for the −741 site, Forward—TTTGGAAGCTTGCAGTCCTT, Reverse—CCGGCTGAAACTTTTTCTCC; for the −1321 site, Forward—CCCAGTTGCTCAAATGGAGT, Reverse—AGGTCGGTGTTTGGTGAGAC. After PCR, 10 uL from a 50 uL reaction were run on a 2% agarose gel to confirm amplification. The remaining PCR reaction was purified using the Qiagen QIAquick PCR Purification Kit, quantified and 10 ng together with 25 pmol of either the Forward or the Reverse primer were submitted to Genewiz (USA) for Sanger sequencing.

### 2.6. RNA Sequencing (RNA-Seq), Real-Time RT-PCR and Differential Expression Analysis

RNA-seq was performed on RNA from 8 fresh-frozen SCCOHT tumors, on a HiSeq2000 sequencer, using paired-end 2 × 76 bp stranded mode. Raw reads were mapped against human genome (hg19) with the STAR (v2.3.0) 2-pass method [19] and potential duplicates were marked using Picard tools (http://picard.sourceforge.net/). Remaining reads were split into exon segments and STAR mapping qualities were reassigned in order to fit GATK (v3.2-2) Indel Realignment requirements [11,20,21]. After local realignment around indels, a base quality score recalibration (BQSR) process was applied, and the variant calling step was done with HaplotypeCaller in RNA-seq mode. Finally, the raw variants list obtained above was filtered on a Phred-scaled p-value using Fisher’s exact test to detect strand bias (FS > 30.0) and Variant Confidence/Quality by Depth (QD < 2.0) values. RNA-seq data were also used to identify variants following Broad Institute Best Practices.

Differential RNA-seq gene expression analysis between SCCOHT samples and benign ovarian tissue from the GTEx dataset was performed using rank-normalized expression values and is detailed in Appendix A.

Quantitative real-time polymerase chain reaction (real-time RT-PCR) analyses to assess the expression levels of *SMARCA2* were performed on RNA from 8 fresh-frozen SCCOHT tumors and from the BIN-67 cell line, as detailed in Appendix A.

### 2.7. Immunohistochemistry

*SMARCA4* (BRG1) and *SMARCA2* (BRM) protein expression was assessed by immunohistochemistry (IHC) using the anti-BRG1 (Santa Cruz, sc-10768) and anti-BRM (Abcam, ab15597) antibodies at dilutions of 1/200 and 1/50, respectively. After paraffin removal and hydration, slides were immersed in 10 mM citrate buffer pH 6 for 30 min for antigen retrieval, incubated with primary antibody for one hour at room temperature, washed and incubated with biotinylated secondary antibody for 30 min at room temperature. Streptavidin-biotin amplification (VECTASTAIN Elite ABC Kit) was then performed for 30 min, followed by peroxidase/diaminobenzidine substrate chromogenic reaction. IHC for SOX2 was performed using a Bond III automated immunostainer and the Bond Polymer Refine detection system (Leica Microsystems, IL, USA). Slides were deparaffinized and heat-mediated antigen retrieval was performed using the Bond Epitope Retrieval 2 solution at pH 9 (H2). The anti-SOX2 antibody clone D6D9 (Cell Signaling Technology) was used at 1/100 dilution.

### 2.8. Cell Culture and Viability Assays

These methods are available in the Appendix A section.

## 3. Results

### 3.1. Clinical Data and Mutational Profiles of SCCOHT: A General Overview

The available clinical data are summarized in Appendix A. The cohort comprised eight patients with a mean age at diagnosis of 31 years (range, 14–40), all of whom were diagnosed with stage III–IV tumors (Figure 1A). Three patients had received chemotherapy prior to sample collection. After a median follow-up of 10 months (range, 3–36 months), seven patients died of disease, while one patient achieved remission and remained disease-free at 36 months follow-up. Histomorphology was reviewed by an expert pathologist (C.G.) and was confirmed to be consistent with SCCOHT for all cases (Figure 1B, Appendix A).

WES was conducted on six tumor-normal pairs. The mean depth of coverage was 109X, with at least 98% of the targeted exome covered by at least 10 reads and 95% showing a read quality score (QC) ≥30 (Appendix A).

The tumor mutation burden (TMB), calculated as specified in the Methods section, was low, with a median of 5.60 mutations/Mb (mean, 5.43 mutations/Mb; range: 3.56–6.42). Very few genes showed somatic-only mutations in more than one sample (Appendix A). These included: *SMARCA4* (three cases, variants in coding regions were predicted to be deleterious and detailed hereafter); *HMCN2* (three cases, missense variants were predicted as benign by Polyphen-2); *ADGRV1* (two cases—one stop gain and one missense variant—both heterozygous), *FANCD2* (two cases, splice region variant predicted to be of low functional impact by the SnpEff tool), and *LRRK2* (two cases, intronic variants).

In order to also account for alterations that may be related to an LOH event, we performed a second analysis using the following criteria: mutant allele frequency higher in tumor than in normal tissue; somatic *p*-value < 0.001 (Fisher’s exact test); and location in a coding region. Using these cut-offs, 500 variants in 335 genes were retained. Among those, fourteen genes were altered in at least 50% of cases (“recurrently altered” genes, Figure 1D,E, Appendix A); importantly, all of these genes were located in 19p13.2-3 and all variants were detected at high allelic frequencies (mean variant allele fraction: 0.88, range: 0.74–0.93), suggesting a recurrent loss-of-heterozygosity (LOH) event in 19p13.2-3. The majority (56/64) of these variants were known polymorphisms (variant frequency in the general population ≥1%) (Figure 1D, Appendix A) and 45/64 were classified as benign by the Polyphen-2 classifier. Variants that were not polymorphisms and that were classified as potentially or probably damaging, or for which functional prediction scores were not available (Figure 1D) included: four variants in *SMARCA4* in five patients (described in detail hereafter; Figure 1E), the p.G223V variant in *PLK5* (one patient), the p.R220H variant in *ACTL9* (one patient) and the c.4208delT frameshift in *ABCA7* (one patient); all of these variants were Sanger-verified (Appendix A).

Lastly, no mutations in the following cancer-related genes were observed in any of the SCCOHT tumors, even at low allelic fractions: *TP53*, *KRAS*, *PIK3CA*, *PTEN*, *BRAF*, *EGFR*, *AKT1*, *CDKN1A* (p21) or *ERBB2*.

### 3.2. Inactivating SMARCA4 Mutations and Related Findings

In line with previous studies, *SMARCA4* (Brahma-related gene 1, BRG1) was mutated by WES in 5/6 (83%) SCCOHT samples in our series (Figure 1C,D). The mean allelic fraction was high (0.86) and consistent with homozygous alterations. The encountered *SMARCA4* mutations were p.N774 frameshift, c.3216-1G>T (splice), p.R1077* stop gain (this mutation was identical in two patients) and p.K1081E (predicted as deleterious by the Polyphen-2 classifier). These variants have been previously reported in Le Loarer et al. [22], as part of the control cohort. All mutations occurred upstream of the SNF2-ATP coupling domain and the bromodomain, suggesting that the functional impact would be a loss of protein expression or function (Figure 1C). In line with those genomic findings, we observed loss of SMARCA4 (BRG1) protein expression by IHC in all cases that displayed *SMARCA4* mutations (Figure 2A).

One tumor (IGR-03) diagnosed as SCCOHT did not exhibit a *SMARCA4* mutation but instead, harbored concomitant and potentially biallelic loss-of-function alterations in two other SWI/SNF genes: *ARID1A* (two frameshifts—p.Q555fs and p.T1004fs) and *ARID1B* (stop gained R1944*). Consequently, SMARCA4 (BRG1) protein expression in this tumor was retained by IHC (Figure 2A).

Of note, one case (IGR-01) also showed a p.Arg635* stop gain in the *SMARCA1* gene, in addition to a deleterious *SMARCA4* mutation.

### 3.3. SMARCA2 Loss of Expression in SCCOHT

Recent studies have shown that in addition to *SMARCA4* inactivation, SCCOHT exhibit a loss of expression of the *SMARCA2* paralog [23]. In our cohort, all *SMARCA4*-mutated SCCOHT (n = 5) showed low/absent *SMARCA2* transcript levels by real-time RT-PCR and complete absence of the SMARCA2 (BRM) protein by IHC (Figure 2A,B), in keeping with previous studies. Combined loss of *SMARCA4* and *SMARCA2* expression in SCCOHT was also confirmed in our extended series of 33 FFPE SCCOHT, as we recently reported [24]. No *SMARCA2* loss-of-function mutations or deletions were found by WES. In the one *SMARCA4* wild-type tumor (IGR-03) which showed concomitant *ARID1A* and *ARID1B* mutations, *SMARCA2* expression was higher at the mRNA level (real-time RT-PCR) than in *SMARCA4* mutated samples, and interpreted as ambiguous/low at the protein level (IHC).

We also validated loss of *SMARCA4* and *SMARCA2* protein expression in two SCCOHT cell lines, BIN-67, and SCCOHT-1, by immunoblotting (Figure 2C). Notably, we observed that expression of other SWI/SNF subunits was retained, in line with a recent study by Pan et al. characterizing the presence of a residual SWI/SNF complex with altered functions in SCCOHT tumor cells [25].

The existence of homozygous insertional polymorphisms of the *SMARCA2* promoter, located −741 bp and −1321 bp from the transcription start site, has previously been linked to loss of *SMARCA2* expression in lung cancer [26]. Thus, we performed Sanger sequencing of the −741 and −1321 promoter sites in eight SCCOHT tumor samples and in two SCCOHT cell lines (BIN-67 and SCCOHT-1). One tumor and one cell line (SCCOHT-1) were homozygous for the −741 polymorphism, another tumor was homozygous for the −1321-promoter site polymorphism, and all other cases displayed a heterozygous −741 and −1321 polymorphism site status (Figure 2D). Overall, we concluded that a homozygous polymorphism site status in the *SMARCA2* promoter, previously described in lung cancer, was not a feature of SCCOHT.

Concomitant loss of *SMARCA4* and *SMARCA2* expression is also a feature of *SMARCA4*-deficient thoracic sarcomas (*SMARCA4*-DTS) and of a subset of malignant rhabdoid tumors (MRTs) [22,27,28]. *SMARCA4*-DTS have been reported to consistently show strong expression of the neural stem cell transcription factor SOX2 [27]. To verify whether the same was true for SCCOHT, 10 FFPE SCCOHT tumors were tested by IHC. Six showed no SOX2 staining (Figure 2E), two showed scattered positive cells and only two showed focal staining (<10% of tumor surface).

### 3.4. Validation of the p.G223V PLK5 Variant in a Larger Series of SCCOHT Samples

*PLK5* is the most recently described member of the Polo-Like Kinase family (PLK) family and has been implicated in involved in DNA damage response and cell cycle checkpoint control [29]. Given the presence of *PLK5* variants in a subset of SCCOHT detected by WES, the potentially damaging p.G223V variant was chosen for further Sanger validation in an extended cohort of 33 FFPE SCCOHT tumors (Appendix A). Overall, this mutation was detected in 3/33 (9%) of SCCOHT, suggesting that although it may be present in SCCOHT, it is not a highly recurrent event.

### 3.5. Somatic Copy Number Alterations (SCNAs) in SCCOHT

Eight fresh-frozen tumors were available for aCGH analysis. As shown in Figure 3B, the aCGH profiles of seven of the eight tumors showed remarkable genomic stability, with few SCNAs. The one tumor exhibiting genomic instability (IGR-07) harbored a loss of *BRCA2*, interpreted as heterozygous [log2 (ratio) = −0.4]. No SCNA was common to ≥50% of tumors, however, 16 recurrent gains were shared by at least three of the eight tumors (Table 1). Four of these genes showed a log2 (ratio) > 2.3, which represents a five-fold increase in copy number, suggestive of amplification: *SHMT2*, *NDUFA4L2*, *LRP1* and *NXPH4*.

### 3.6. Copy-Neutral Loss-of-Heterozygosity (CN-LOH) at the 19p13.2-3 Locus

WES revealed recurrent loss-of-heterozygosity (LOH) at the 19p13.2-3 region in five of six tumors and nominated the smallest common LOH region as Chr19:373.916-11.465.316 (Figure 3A). We mapped this “common LOH-region” by aCGH and, as shown in Figure 3B,C, confirmed that no copy number losses were present in this region, thus supporting the presence of a recurrent copy neutral-LOH (CN-LOH) event (Figure 3D).

### 3.7. Gene Expression Profiles of SCCOHT

To compare gene expression data in SCCOHT with our genomic findings, we conducted RNAseq-based differential expression analysis comparing six *SMARCA4*-mutated SCCOHT cases from our cohort and five samples of benign ovarian tissue from premenopausal women from the GTEx dataset. The analysis was conducted on rank-normalized gene expression values to reduce batch effect (Appendix A). As expected, *SMARCA2* expression was significantly lower in SCCOHT than in benign ovarian tissue, with mean rank-normalized expression values of 0.50 vs. 0.95, log2 fold change = −0.92 and padj = 0.015. No statistically significant difference in *SMARCA4* expression was detected, possibly due to low *SMARCA4* expression in two of the benign ovarian samples.

None of the four genes that showed recurrent amplifications in SCCOHT (*NDUFA4L2*, *SHMT2*, *NXPH4*, *LRP*) was significantly overexpressed in this analysis. *PLK5* showed very low expression values in both groups (mean rank-normalized expression values of 0.18 and 0.17, respectively). SCCOHT also did not show significant overexpression of *SOX2* (consistently with our IHC data) or of *PTHLH* (which encodes Parathyroid Hormone-Related Protein, previously postulated to cause hypercalcemia in a subset of SCCOHT).

Differential expression analysis also allowed the nomination of some genes potentially overexpressed in SCCOHT. Overall, ~1900 significantly differentially expressed genes showed log2 fold change > 1 or < −1 and padj < 0.05 (Appendix A). The most significantly overexpressed genes (top 100) included: cancer-testis antigens (e.g., *MAGEA4,* which was also the most significantly overexpressed gene, *MAGEA9, DSCR8, SYCE3*); the *AURKB* gene, encoding an Aurora kinase involved in mitotic progression; the tyrosine kinase receptor gene *ERBB4* (HER4); genes encoding metalloproteinases (e.g., *MMP10*, *MMP9*, *MMP1*); genes related to neural development (e.g., *NCAM2*, *NTS*, *ATCAY*, *CBLN2*); embryonic stem cell genes (*CLDN6*, which encodes an embryonic cell junction protein); and germ cell markers (*SALL4*) (Figure 4A, Appendix A). Conversely, the expression of some genes known to be highly expressed in benign ovarian tissue was significantly lower in SCCOHT (e.g., *INHA*, *FOXL2*, *AMHR2*). Gene Set Enrichment Analysis (GSEA) (Figure 4B, Appendix A) showed that gene sets significantly enriched in SCCOHT included those related to E2F targets and cell cycle progression, DNA repair, activation of oncogenic pathways (KRAS, MYC, mTORC1), as well as gene sets related to *SMARCB1* (SNF5) knockdown, consistent with a deregulated SWI/SNF complex.

### 3.8. Genomic and Transcriptomic Profiles of Chemotherapy-Naïve Versus Chemotherapy-Exposed SCCOHT

SCCOHT are characterized by initial chemosensitivity, but almost invariably relapse. Thus, we compared mutation profiles in the three treatment-naïve and three chemotherapy-exposed tumors, to determine whether some alterations were enriched in post-treatment samples. The chemotherapy regimens received by patients IGR01, IGR04 and IGR06 prior to sample collection/surgery were VIP/Doxorubicin, BEP/ PAVEP and EP/PAVEP, respectively (Appendix A).

Tumor mutation burden was not significantly higher in the three post-chemotherapy samples than in the three chemo-naïve ones: mean, 4.98 non-synonymous mutations/Mb (range: 3.56–6.42) and mean, 5.87 mutations/Mb (range, 5.21–6.42), respectively (*p* = 0.38, unpaired t-test). Among somatic-only mutations (i.e., mutations not imputable to an LOH event), two genes were altered in at least two post-chemotherapy samples, but not in chemo-naïve samples: *ADGRV1* (IGR-01 and IGR-06—one stop gain and one missense alteration) and *FANCD2* (IGR-04 and IGR-06—splice region variant predicted to be of low functional impact by the SnpEff tool). No SCNAs were differentially detected in the chemotherapy-exposed versus chemotherapy-naïve tumors (data not shown).

RNAseq-based differential expression analysis between the chemotherapy-exposed and the chemotherapy-naïve samples did not identify any significantly differentially expressed genes (data not shown). However, when analyzing ranked genes in their totality, GSEA analysis revealed several gene sets with significant positive or negative enrichment (Appendix A). Among these, there was a positive enrichment of the gene set reflecting genes upregulated upon overexpression of Eukaryotic Translation Initiation Factor 4E (eEIF4E), a positive enrichment of gene sets downregulated upon KRAS overexpression, and a negative enrichment of genes upregulated upon KRAS overexpression (Figure 4C).

### 3.9. Epigenetic Vulnerabilities of SCCOHT Associated with SWI/SNF Deregulation

In a recent study, Pan et al. have shown that the loss of catalytic SWI/SNF activity in SCCOHT largely alters SWI/SNF functions as an epigenetic regulator [25]. To assess the putative sensitivity of SCCOHT to currently available epigenetic treatments, cell lines with differing *SMARCA4* genotypes were treated with the histone deacetylase inhibitor trichostatin A (TSA) and the DNA methyltransferase inhibitor 5′-dAZAC. The SCCOHT cell line BIN-67, which harbors an inactivating *SMARCA4* mutation and shows complete loss of *SMARCA2* expression (Figure 2C), was exquisitely sensitive to 5′-dAZAC and TSA at sub-nanomolar concentrations (Figure 5A,B). Conversely, the H1299 lung cancer cell line, which carries a *SMARCA4* mutation, but shows retained *SMARCA2* expression (*SMARCA4-/SMARCA2+*), and the ovarian high-grade endometrioid adenocarcinoma cell line SKOV3 (*SMARCA4+/SMARCA2+*) were completely resistant to 5′-dAZAC and 100-fold less sensitive to TSA than BIN-67 (Figure 5A,B).

Many studies suggest an antagonistic relationship between the SWI/SNF complex and Polycomb proteins, such as Enhancer of Zest 2 (EZH2) [30]. Combined loss of *SMARCA4* and *SMARCA2* in SCCOHT cell lines may induce an oncogenic dependency on EZH2 activation [31] and confer extreme sensitivity to EZH2 inhibitors in vitro and in vivo [32]. Based on this rationale, we enrolled a patient with *SMARCA4*-mutated SCCOHT in a phase I trial of tazemetostat (EPZ-6438), a highly selective EZH2 inhibitor [33]. This 25-year-old patient initially presented with stage IV SCCOHT treated with surgery, combination platinum-based chemotherapy followed by high dose consolidation and autologous stem cell rescue (Figure 5C). Unfortunately, she relapsed within eight weeks and was, therefore, enrolled in the EZH2 inhibitor clinical trial. She presented partial response (RECIST 1.1) after four months of treatment and remained progression-free for eight months. Although the clinical benefit was relatively short, the degree of response in this patient with highly chemo-resistant disease supports further investigation of epigenetic strategies in SCCOHT.

## 4. Discussion

SCCOHT are rare tumors that occur in young women and their prognosis remains poor, despite aggressive multimodal therapy. We present an integrated molecular characterization of additional cases of SCCOHT from an independent cohort.

Intriguingly, our findings and the previously published data show that these aggressive tumors carry a diploid DNA content, which is a rare phenomenon in a highly lethal malignancy [34,35]. In addition, we show that SCCOHT have a very low mutation load (mean, 5.43 mut/Mb) and lack mutations in genes most altered across various cancer types. Collectively, these observations support the hypothesis that SCCOHT are largely driven by epigenetic deregulation and not by genomic instability.

Importantly, our results underscore marked inter- and intra-tumor homogeneity of SCCOHTs. Combined WES and aCGH analysis revealed a recurrent copy-neutral LOH (CN-LOH) at the 19p13.2-3 locus. CN-LOH can account for inactivation of tumor suppressor genes and likely implicates the loss of the normal allele and duplication of the mutated copy. Notably, 19p LOH has previously been detected in SCCOHT by WES [4,36], but our results provide additional evidence for a copy-neutral nature of this event. Of note, telomeric CN-LOH has been linked to meiotic errors occurring during cross-over [37], which could be in line with the postulated germ cell origin of SCCOHT tumors [2,3], although further studies are needed to support this hypothesis. The 19p CN-LOH associated with inactivating *SMARCA4* mutations has also been reported in non-small cell lung cancer [38].

In line with previous studies, we found that *SMARCA4* mutations were present in all but one SCCOHT (5/6) in our series. *SMARCA4* encodes one of the enzymatic (ATP-ase) subunits of mammalian SWI/SNF, a chromatin remodeling complex which directs nucleosomes and modulates gene expression. The importance of SWI/SNF alterations in oncogenesis or tumor progression is being increasingly acknowledged, as alterations in SWI/SNF subunits are found in over 20% of human cancers [36]. Early preclinical studies suggested that *SMARCA4*-mutated tumors (such as non-small cell lung cancers) were critically reliant on the *SMARCA2* paralog [29,39]. Conversely, SCCOHT do not seem amenable to this synthetic lethality strategy, given the complete loss of *SMARCA2* expression demonstrated in our series and in previous studies [22]. This loss of expression is not explained by mutations in the coding sequence of *SMARCA2*. In an effort to explore the underpinnings of *SMARCA2* silencing, we report for the first time that homozygous polymorphisms at the two *SMARCA2* promotor polymorphism sites (−741 bp and −1321 bp), previously linked to *SMARCA2* silencing in cancer [15,40], do not seem to be a recurrent event responsible for *SMARCA2* silencing in SCCOHT. Nevertheless, since most tumors in our study showed a heterozygous promotor polymorphism site status, further explorations are warranted to elucidate whether heterozygous polymorphisms can contribute to *SMARCA2* silencing. In particular, in malignant rhabdoid tumor cell lines, increased binding of epigenetic silencers HDAC9 and MEF2D at *SMARCA2* promoter sites has been associated with such heterozygous polymorphisms [41].

One case (IGR-03) from our series exhibited concomitant inactivating mutations in *ARID1A* and *ARID1B*, two paralog DNA-binding subunits of SWI/SNF, but did not show *SMARCA4* mutations. This tumor was also the only case in which 19p CN-LOH was not present. Concomitant *ARID1A/B* alterations occur in ~25% of dedifferentiated endometrial and ovarian carcinomas [42]. While case IGR-03 could illustrate the challenges of differential diagnosis between SCCOHT and dedifferentiated ovarian carcinoma, another possibility is the existence of a molecular and morphologic overlap between those two entities, both of which are characterized by a poorly differentiated, aggressive tumor and a critically deregulated SWI/SNF complex.

Of note, one case (IGR-01) showed a p.Arg635* stop gain in the *SMARCA1* gene in addition to a deleterious *SMARCA4* mutation. *SMARCA1* encodes the ATP-ase of another chromatin remodeling complex, ISWI, and is located on the X chromosome, suggesting that this alteration, which was seen at an allele frequency of 0.37, could potentially carry a deleterious impact.

As a complement to the genomic findings, we also show for the first time that SCCOHT are not characterized by SOX2 overexpression, contrary to another aggressive *SMARCA4/SMARCA2* double-negative malignancy—*SMARCA4*-deficient thoracic sarcoma (*SMARCA4*-DTS). This emphasizes the existence of biological differences between *SMARCA4*-DTS and SCCOHT, in addition to previously described discrepancies, such as higher genomic instability in *SMARCA4*-DTS [26], and could have potential implications in diagnostic pathology.

In addition to *SMARCA4* loss-of-function alterations, a few variants were seen in genes other than *SMARCA4*, all localized in the 19p13.2-3 locus and subject to the CN-LOH event, for which a functional impact could not be ruled out. In particular, the *PLK5* p.G223V variant, predicted as potentially damaging by the Polyphen-2 classifier, was found in 3/33 SCCOHT samples of the extended cohort. The protein kinase domain of *PLK5* is truncated in humans compared to mice, but the residual protein containing the polo-box binding domain may act as a stress inducible tumor suppressor regulating G1 arrest [29]. Nevertheless, the relevance of these variants remains to be validated functionally.

Differential expression analysis comparing SCCOHT and benign ovarian tissue allowed to nominate several genes potentially overexpressed in SCCOHT. Nevertheless, it must be kept in mind that the bulk benign ovarian tissue used as the control in this analysis does not represent the exact cell origin of SCCOHT, which remains unknown. Among other findings, we observed significant overexpression of some putative therapeutic targets. Cancer-testis antigens have been proposed as targets for immunotherapy approaches and Melanoma-associated antigen 4 (*MAGEA4*), which was the most highly overexpressed gene in SCCOHT, is currently being investigated as a TCR-engineered T-cell target (NCT03247309). The *AURKB* gene encodes Aurora B kinase, implicated in mitotic progression, and may be targeted by pharmaceutical inhibitors (e.g., GSK1070916). The overexpression of receptor tyrosine kinase genes, such as *ERBB4* (HER4), could potentially be in line with a recent study showing marked vulnerability of SCCOHT cells to multi-kinase inhibition [43]. Intriguingly, SCCOHT also showed expression of neural differentiation markers and embryonic stem cell markers, in keeping with what has previously been reported in malignant rhabdoid tumors [44]. Some of these markers could represent treatment opportunities, such as the embryonic cell junction protein Claudin-6 (*CLDN6*), against which monoclonal antibodies were recently part of a clinical trial in ovarian cancers (NCT02054351). Nevertheless, further studies are needed to confirm overexpression of these putative treatment targets at the protein level and to validate their functional relevance in SCCOHT. 

To further explore putative therapeutic approaches, we found that the SCCOHT cell line BIN67 was exquisitely sensitive to TSA and 5′-dAZAC, while cells with a *SMARCA4* mutation and retained *SMARCA2* expression were resistant to these epigenetic therapies. In addition, we describe a clinically meaningful response to single agent EZH2 inhibitor in a patient with SCCOHT, in keeping with what has previously been demonstrated in vitro and in vivo [32]. Collectively, data from our series and from previous studies suggest that SCCOHT tumors characterized by the loss of both SWI/SNF catalytic subunits may be sensitive to treatment with HDAC, DNA methyltransferase and/or EZH2 inhibitors, and that such strategies merit further investigation in this lethal disease.

Since SCCOHT often display initial chemosensitivity, but subsequently show rapid progression, we compared the genomics of treatment-naïve (n = 3) versus chemotherapy-treated tumors (n = 3), to uncover candidate resistance genes. Critically, neither the tumor mutation burden nor the somatic copy number alterations were significantly increased in post-chemotherapy samples. The only alterations seen in more than one post-chemotherapy sample and not in chemotherapy-naïve samples were *ADGRV1* (one stop gain and one missense alteration) and *FANCD2* (splice region variants). Variants in *ADGRV1*, which encodes adhesion G protein-coupled receptor 1, were potentially heterozygous, and the implication of this protein in cancer progression is unclear. Variants in *FANCD2* were predicted to be of low functional impact by the SnpEff tool. Although differential expression analysis did not reveal any specific genes significantly deregulated between the two groups, GSEA performed on the totality of ranked genes nominated several deregulated gene expression programs, including an enrichment of genes related to eIF4E upregulation and a putative downregulation of the KRAS pathway. Further studies comparing paired samples from the same patient before and after treatment are needed to elucidate molecular underpinnings of treatment resistance in SCCOHT. Nevertheless, our data suggest that it may rely on other mechanisms than acquiring drug resistance mutations, contrary to what has been described in other tumor types.

## 5. Conclusions

SCCOHT are unique tumors. Despite their aggressive clinical course, they display genomic stability, low mutation load, few SCNAs, and a remarkably homogeneous genomic profile. They are almost universally characterized by a 19p CN-LOH, loss-of-function mutations in *SMARCA4* and concomitant loss of *SMARCA2* expression. SCCOHT do not seem to acquire additional mutations after exposure to chemotherapy. Some additional molecular alterations reported herein could be further explored as therapeutic opportunities, such as the overexpression of putative therapeutic targets (e.g., *MAGEA4*, *AURKB* or *CLDN6*). Our preliminary in vitro data and the reported patient case also support the view that SCCOHT are sensitive to epigenetic modulators, such as HDAC, DNMT and EZH2 inhibitors, in line with other studies and with early results of phase I trials, and warrant further exploration of epigenetic treatment strategies in this lethal disease.

## Figures and Tables

**Figure 1 cells-09-01496-f001:**
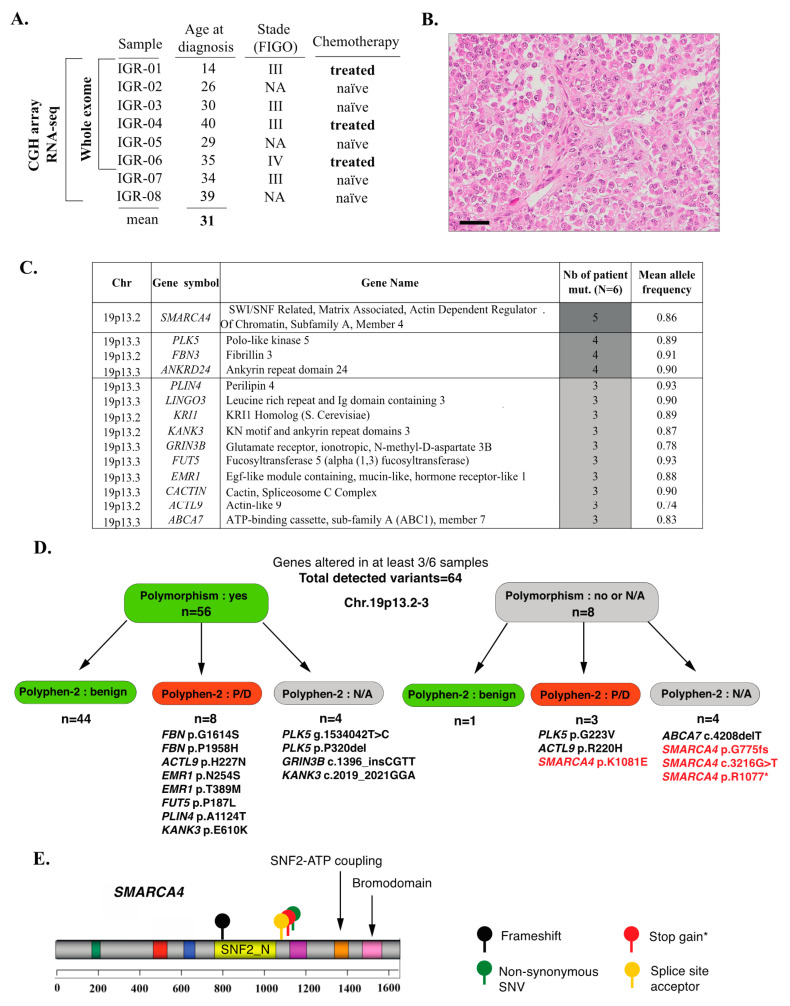
An overview of mutational profiles of SCCOHT. (**A**) Clinical characteristics of the cohort and tests performed. (**B**) Representative histopathology of a SCCOHT case from this cohort (IGR-04), including rhabdoid features; hematoxylin-eosin-saffron, scale bar: 50 μm. (**C**) Combined analysis of somatic-only and LOH-related alterations: an overview of the 14 genes altered in at least 50% of samples. (**D**) Breakdown of variants detected in the 14 recurrently altered genes, including classification as known polymorphisms (Genome Aggregation Database v.2.1.1) and Polyphen-2 functional prediction scores. N/A: not available. P/D: possibly or probably damaging. The variants for which functional impact cannot be ruled out are explicitly listed. (**E**) Type and localization of the mutations found by WES in the *SMARCA4* gene; * indicates that this identical mutation was found in two independent patients.

**Figure 2 cells-09-01496-f002:**
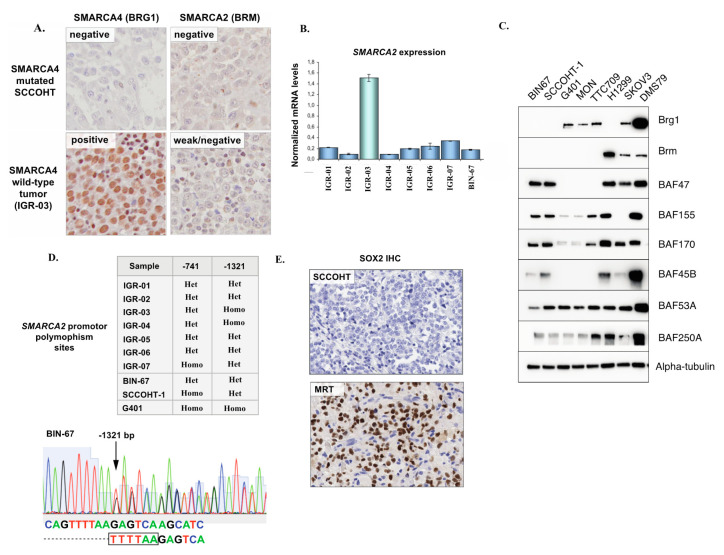
*SMARCA4* and *SMARCA2* expression in SCCOHT. (**A**) Representative *SMARCA4* and *SMARCA2* immunohistochemistry in a *SMARCA4* mutated SCCOHT and in the one *SMARCA4* wild-type. Tumor harboring concomitant *ARID1A* and *ARID1B* mutations (IGR-03). (**B**) Real-time RT-PCR for *SMARCA2* in patient tumor samples from this study and in a SCCOHT cell line (BIN-67); expression levels are normalized to three housekeeping genes (*YWHAZ/GUSB/HPRT1*). (**C**) Western blot showing expression of several SWI/SNF subunits in SCCOHT cell lines (BIN-67, SCCOHT-1) compared to MRT (G401, MON, TTC709), *SMARCA4*-mutated lung cancer (H1299), high-grade endometrioid adenocarcinoma of the ovary (SKOV3) and neuroendocrine small cell lung cancer (DMS79) cell lines. (**D**) Results of Sanger sequencing of the *SMARCA2* promoter insertional polymorphism sites, and an example of a heterozygous polymorphism status (−1321 site) in BIN-67 cells. (**E**) Representative IHC for SOX2 in SCCOHT and a positive control (SOX2-positive MRT) in patient FFPE tumor samples.

**Figure 3 cells-09-01496-f003:**
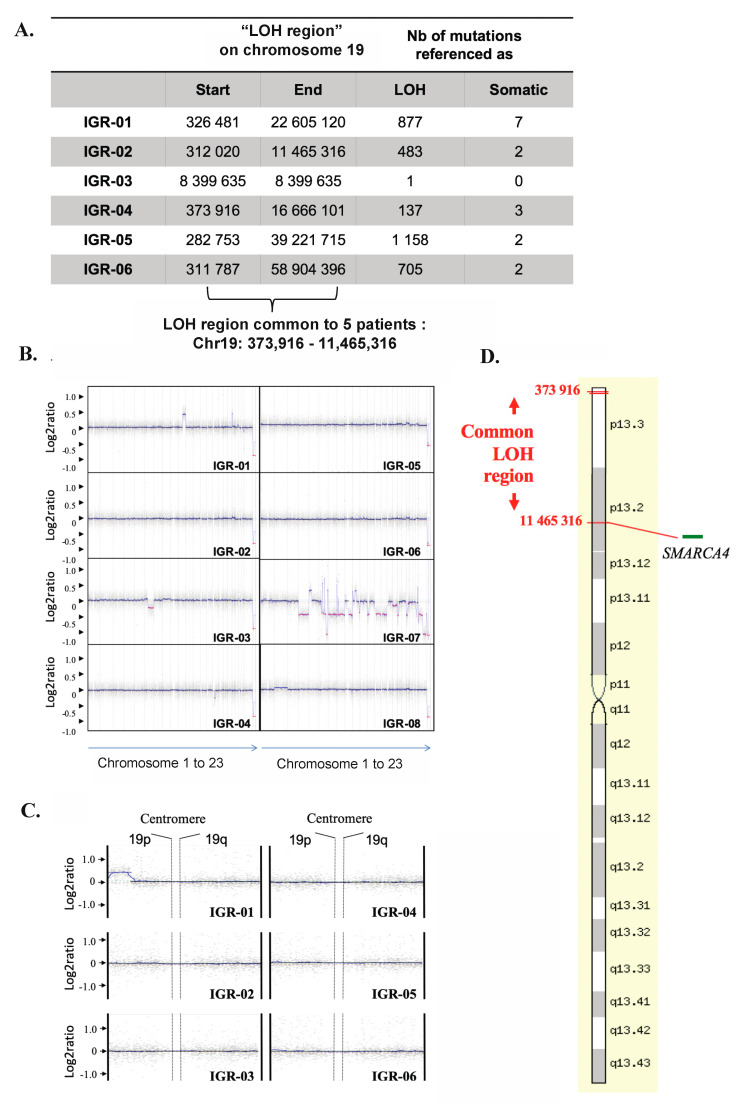
SCCOHT demonstrate remarkable genomic stability and recurrent 19p CN-LOH. (**A**) LOH regions obtained by WES in each tumor identifies a common “LOH region” on chromosome 19 for all SCCOHTs except IGR03: Chr19:373916-11465316. (**B**) CGH array profiles for each patient. (**C**) Zoom on 19p in all tumors fails to show a heterozygous copy number loss, thus suggestive of copy neutral LOH. (**D**) Artificial representation of the “common LOH region” on chromosome 19 in tumors (source: http://www.genecards.org).

**Figure 4 cells-09-01496-f004:**
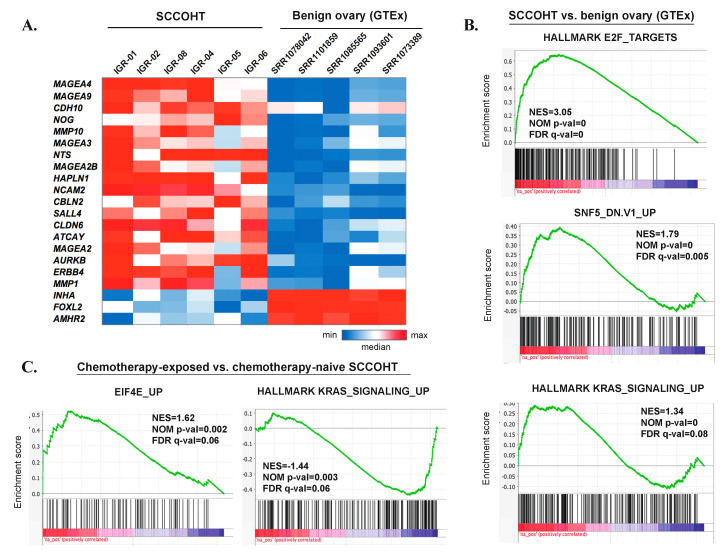
An overview of transcriptomic profiles of SCCOHT. (**A**) Graphic heatmap representation of rank-normalized expression values for selected, most significantly deregulated genes in the differential expression analysis between SCCOHT and benign ovarian tissue (GTEx). (**B**) Selected GSEA results for the differential expression analysis between SCCOHT and benign ovarian tissue (GTEx). (**C**) Selected GSEA results for the differential expression analysis between chemotherapy-exposed SCCOHT samples (IGR-01, IGR-04, IGR-06) and chemotherapy-naïve samples (IGR-02, IGR-05, IGR-08).

**Figure 5 cells-09-01496-f005:**
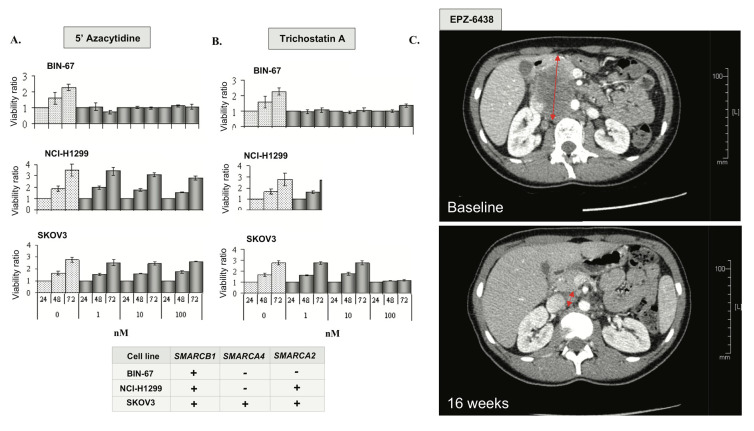
Epigenetic vulnerabilities in SCCOHT A, B. Anti-proliferative effects of 5′-AZAC (**A**) and TSA (**B**); − designates protein loss or loss-of-function mutation and/or loss of expression; + designates absence of mutation (wild-type status) and retained expression. (**C**) Rapid clinical response in *SMARCA4*-mutated SCCOHT treated with the EZH2 inhibitor EPZ-6438. A CT scan of the tumor at baseline and after four months of EPZ-6438 treatment with 70% decrease in tumor volume (RECIST 1.1).

**Table 1 cells-09-01496-t001:** Recurrent Gains Shared by at Least three of eight Tumors. Aberrant SCNAs Were Defined as log2 (ratio) < −1 or > 1.

Localization	Gene Symbol	Description	Mean Log2Ratio
**12q12-q14**	SHMT2	Serine hydroxymethyltransferase 2 (mitochondrial)	2.33
**12q13.3**	NDUFA4L2	NADH dehydrogenase (ubiquinone) 1 alpha subcomplex, 4-like 2	2.33
**12q13.3**	NXPH4	Neurexophilin 4	2.33
**12q13.3**	LRP1	Low density lipoprotein receptor-related protein 1	2.33
**14q32.2**	BEGAIN	Brain-enriched guanylate kinase-associated	2.23
**14q32.2**	LINC00523	Long intergenic non-protein coding RNA 523	2.23
**16q24**	CBFA2T3	Core-binding factor, runt domain, alpha subunit 2; translocated to, 3	1.40
**16q24**	APRT	Adenine phosphoribosyltransferase	1.40
**16q24.3**	ACSF3	Acyl-CoA Synthetase Family Member 3	1.40
**16q24.3**	CTU2	Cytosolic thiouridylase subunit 2 homolog (S. pombe)	1.40
**16q24.3**	GALNS	Galactosamine (N-acetyl)-6-sulfate sulfatase	1.40
**16q24.3**	MIR4722	MicroRNA 4722	1.40
**16q24.3**	PABPN1L	Poly(A) binding protein, nuclear 1-like (cytoplasmic)	1.40
**16q24.3**	CDT1	Chromatin licensing and DNA replication factor 1	1.40
**16q24.3**	PIEZO1	Piezo-type mechanosensitive ion channel component 1	1.40
**16q24.3**	TRAPPC2L	Trafficking protein particle complex 2-like	1.40

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
