# Peer review of "Small Cell Carcinoma of the Ovary, Hypercalcemic Type (SCCOHT) beyond *SMARCA4* Mutations: A Comprehensive Genomic Analysis"

_cells, 2020, doi:10.3390/cells9061496_

Round 1

Reviewer 1 Report

Aurélie et al. report in their revised and newly submitted manuscript about the characterization of genomic aberrations in Small cell carcinoma of the ovary, hypercalcemic type (SCCOHT). This tumor type is very rare but has a rather low survival rate thus representing a target for both basic and clinical research to improve the prognosis of affected patients. The knowledge about genomic aberrations besides SMARCA4 mutations is limited. Therefore the clinical need and the limited knowledge justify the presented work. The authors conducted whole exome sequencing, RNA sequencing and CGH analyses to identify aberrations. Validations were partially done by targeted analyses using other samples or functional studies with exemplary cell cultures. The authors responded meaningful to the comments and changed their manuscript. However, following remarks should be addressed by the authors.

Major remarks

  1. Albeit the authors adapted their manuscript partially and the reviewer agrees with the report of variants enriched in tumors by the 19pCN-LOH he still sees some limitations which should be addressed and at least mentioned in the manuscript. (1) Were the enriched SNVs counted as mutation for the mutation rate calculation? To calculate the mutation rate only tumor-specific de-novo mutations should be counted (e.g. ACTL9 p.R220H; SMARCA4 p.K1081E). (2) The presented analysis of chemo-naïve and treated patients is biased by the LOH (see point 2). (3) The analysis of PLK5 sequences of the enlarged cohort reports mutation data (Suppl. Tab. S5) which is very likely biased and not correct (see point 3).
  2. The authors analyzed tumor samples form chemotherapy-naïve and treated patients. Albeit only a low number or patients are included these data are interesting and may lead to new hypothesis. To increase the information the authors should add data about the time span between chemotherapy and surgery and the type of applied chemotherapy. The authors state that both groups of tumors do not contain a significantly different mutational burden (line 325ff). How did they test for significance? Albeit the difference is not large (0.94 vs. 0.31 mutations per Mb) the 3 tumors from treated patients show the highest mutation frequency. This may result in an even lower mean mutation rate for chemo-naïve SCCOHT. The authors compared both groups for the identified genes (excluding PLIN4, LINGO3 and ACTL9?) with recurrently enriched SNVs (Suppl. Tab. S4). These data are unlikely to inform anything about chemotherapy induced genomic changes because only the recurrent SNVs are analyzed and most of them are enriched by the LOH only. Association between polymorphisms of the 19p region affected by the LOH (itself being highly recurrent) and present in normal blood occur very likely by chance (as mentioned in the discussion) and should not be included in the results section but only discussed. However, the authors should reanalyze their exome-wide mutation data to identify de-novo mutations specific for the treated tumors. Likewise, it would be interesting to compare the genome-wide RNA expression between chemo-naïve and treated tumors. These additional analyses should be conducted only if the 3 treated patients obtained the same/similar chemotherapy.
  3. Supplementary table S5 should be changed because shown data (except G223V) represent enriched SNVs not mutations (if compared to population data Tab. S2). Did the authors analyze normal tissue/blood from these patients to identify tumor-specific mutations? Does Tab. S5 include patients from the genome analyses (because of identical labels)?

Minor remarks

  1. It would be interesting to include information about patient-specific sequence variation data (which patient harbours which SNV/mutation).
  2. The material and method section contains several information which are not included in the manuscript, please check (OVCAR3, inhibitors…)
  3. Lines 325-333 the references to tables and figures should be checked.
  4. The SKOV3 cell line is named as serous. This is not correct.
  5. For the cell viability assay the authors should state used cell number per well and include information about the data calculation.
  6. The authors should check if all authors fulfill the requirements for authorship.

Author Response

Reviewer 1

Comments and Suggestions for Authors

Aurélie et al. report in their revised and newly submitted manuscript about the characterization of genomic aberrations in Small cell carcinoma of the ovary, hypercalcemic type (SCCOHT). This tumor type is very rare but has a rather low survival rate thus representing a target for both basic and clinical research to improve the prognosis of affected patients. The knowledge about genomic aberrations besides SMARCA4 mutations is limited. Therefore the clinical need and the limited knowledge justify the presented work. The authors conducted whole exome sequencing, RNA sequencing and CGH analyses to identify aberrations. Validations were partially done by targeted analyses using other samples or functional studies with exemplary cell cultures. The authors responded meaningful to the comments and changed their manuscript. However, following remarks should be addressed by the authors.

Major remarks

  1. Albeit the authors adapted their manuscript partially and the reviewer agrees with the report of variants enriched in tumors by the 19pCN-LOH he still sees some limitations which should be addressed and at least mentioned in the manuscript.

(1) Were the enriched SNVs counted as mutation for the mutation rate calculation? To calculate the mutation rate only tumor-specific de-novo mutations should be counted (e.g. ACTL9 p.R220H; SMARCA4 p.K1081E).

We thank the Reviewer for these comments and we agree that alterations imputable to the CN-LOH should not be part of the TMB calculation. Thus, we performed a new TMB analysis, using non-synonymous and somatic-only mutations located in coding regions and a somatic p-value threshold at <0.05 (instead of the previously used 0.001), in line with the methodology used in other studies (e.g. PMID 22318521, 27322744). Using these criteria, the TMB showed a median of 5.60 mutations/Mb (mean, 5.43 mutations/Mb; range: 3.56-6.42). While these numbers are higher than those mentioned in the previous version of the manuscript (due to the use of a less stringent cut-off for somatic p-value), they remain low and are consistent with the TMB range reported in SCCOHT by another team (PMID: 29102090). These numbers have now been incorporated into the revised version of the manuscript.

(2) The presented analysis of chemo-naïve and treated patients is biased by the LOH (see point 2).

Similarly to Point 1, we excluded LOH events from the comparative analysis of mutations between chemo-exposed and chemo-naïve samples. In particular, the ABCA7 alterations were no longer differentially identified in this new analysis and thus, this gene has been excluded from the Results section and Discussion. Conversely, the current version of the manuscript mentions alterations in ADGRV1 (IGR-01 and IGR-06; one stop gain and one missense alteration, both heterozygous) and FANCD2 (IGR-04 and IGR-06; splice region variant predicted to be of low functional impact).

(3) The analysis of PLK5 sequences of the enlarged cohort reports mutation data (Suppl. Tab. S5) which is very likely biased and not correct (see point 3).

We agree with the Reviewer and we have revisited the PLK5 results. Only the p.G223V variant, considered as potentially deleterious, is discussed in the updated version of the manuscript and in the supplemental material (currently Supplementary table S5).

2. The authors analyzed tumor samples form chemotherapy-naïve and treated patients. Albeit only a low number or patients are included these data are interesting and may lead to new hypothesis. To increase the information the authors should add data about the time span between chemotherapy and surgery and the type of applied chemotherapy.

We thank the Reviewer for this suggestion. To provide more information about this rare disease and to offer more granularity regarding the chemo-exposed and chemo-naïve status of the tumor samples, we collected additional clinical details regarding the eight patients, including the type of chemotherapy received and patient outcome, which are now presented as Supplementary table 1 and referenced in the text. For some patients, information was not available. Unfortunately, we were not able to collect information on the exact time span between chemotherapy and surgery.

The authors state that both groups of tumors do not contain a significantly different mutational burden (line 325ff). How did they test for significance? Albeit the difference is not large (0.94 vs. 0.31 mutations per Mb) the 3 tumors from treated patients show the highest mutation frequency. This may result in an even lower mean mutation rate for chemo-naïve SCCOHT.

This is a very good point. As mentioned in our reply to Point 1, using the new criteria, the TMB showed a median of 5.60 mutations/Mb (mean, 5.43 mutations/Mb; range: 3.56-6.42).

The new comparative analysis between chemotherapy-naïve and chemotherapy-exposed samples did not reveal a statistically significant difference between the two groups: mean, 4.98 non-synonymous mutations/Mb (range: 3.56-6.42) and mean, 5.87 mutations/Mb (range, 5.21-6.42), respectively (p=0.38, unpaired t-test). This result has been updated in the revised version of the manuscript.

The authors compared both groups for the identified genes (excluding PLIN4, LINGO3 and ACTL9?) with recurrently enriched SNVs (Suppl. Tab. S4). These data are unlikely to inform anything about chemotherapy induced genomic changes because only the recurrent SNVs are analyzed and most of them are enriched by the LOH only. Association between polymorphisms of the 19p region affected by the LOH (itself being highly recurrent) and present in normal blood occur very likely by chance (as mentioned in the discussion) and should not be included in the results section but only discussed.

However, the authors should reanalyze their exome-wide mutation data to identify de-novo mutations specific for the treated tumors.

We agree with the Reviewer that the 19pCN-LOH event introduces and important bias in the analysis. In addition, alterations imputable to LOH most likely pre-date chemotherapy, as this genomic event is postulated to be the driver event which also causes SMARCA4 LOH. To address this, we excluded LOH events from the comparative analysis of mutations between chemotherapy-exposed and chemotherapy-naïve samples. Given that ABCA7 alterations were no longer differentially identified in this new analysis, this gene has been excluded from the Results section and Discussion. Conversely, we cite two other alterations that were specifically identified in the new analysis: ADGRV1 (IGR-01 and IGR-06; one stop gain and one missense alteration, both heterozygous) and FANCD2 (IGR-04 and IGR-06; splice region variant predicted to be of low functional impact).

To give the Reader more insight into the presented data, somatic-only SNVs per sample have now been included as Supplementary table S3.

Likewise, it would be interesting to compare the genome-wide RNA expression between chemo-naïve and treated tumors. These additional analyses should be conducted only if the 3 treated patients obtained the same/similar chemotherapy.

As shown in the new Supplementary table S1, the chemotherapy received by the three patients prior to sample collection was somewhat different. Nevertheless, we performed a differential expression analysis between the three chemotherapy-exposed and chemotherapy-naïve samples for completeness. The analysis did not reveal any significantly differentially expressed genes between the two groups. However, when looking at the totality of ranked genes from this analysis, a few gene sets were found to be significantly enriched by GSEA. These results have been added to the revised version of the manuscript (Fig. 4C, Supplementary table S10).

3. Supplementary table S5 should be changed because shown data (except G223V) represent enriched SNVs not mutations (if compared to population data Tab. S2).

We agree with the Reviewer. Only the p.G223V variant has been retained in the current version of the table.

Did the authors analyze normal tissue/blood from these patients to identify tumor-specific mutations?

For the 6 samples for which WES was performed, we did analyze tumor-normal pairs (normal tissue/blood was used as control).

For the validation cohort (Sanger sequencing on DNA extracted from FFPE tumor tissue), normal tissue or blood was not tested, as it was not available and/or a specific patient consent was not available for the validation cohort.

Does Tab. S5 include patients from the genome analyses (because of identical labels)?

Indeed, there is an overlap between these analyses and the patient identifiers remain the same throughout the different analyses. For these patients, this table represents as Sanger validation of the WES data. This specification has now been included in the table (now Supplementary table S6).  

Minor remarks

  1. It would be interesting to include information about patient-specific sequence variation data (which patient harbours which SNV/mutation).

We agree that transparency is important. To give the Reader more insight into the presented data, somatic-only SNVs per sample have now been included as Supplementary table S3.

2. The material and method section contains several information which are not included in the manuscript, please check (OVCAR3, inhibitors…)

We thank the Reviewer for pointing out these inconsistencies. The inhibitors, which are no longer mentioned in the manuscript, have now been removed from Supplementary methods. The OVCAR3 cell line was used in Sanger validation of PLK5 alterations and has therefore been kept in the Supplementary methods.

3. Lines 325-333 the references to tables and figures should be checked

We have verified and amended the references. We hope that the current version of the manuscript is correct.

4. The SKOV3 cell line is named as serous. This is not correct.

We thank the Reviewer for making us aware of this misconception. Indeed, although this cell line is listed as high-grade serous adenocarcinoma across various cell line databases (e.g. ExPASy Bioinformatics Resource Portal) and harbors a TP53 mutation, we have now seen that published molecular analyses have reclassified this cell line as an endometrioid adenocarcinoma (PMID 25230021). This has been corrected in the manuscript.

5. For the cell viability assay the authors should state used cell number per well and include information about the data calculation.

Cells were seeded into 96-well plates at 1.2×104 cells/well (i.e., a suspension of 1.2×105 cells/mL and a 100 mL working volume per well). Viability ratios (using the 0h timepoint as reference) between treated and untreated cells at 72h were compared using a two-way ANOVA test. These specifications have been added to the Supplementary methods document.

6. The authors should check if all authors fulfill the requirements for authorship.

The authors confirm that all collaborators included as co-authors of this manuscript have contributed substantially to this work.

Reviewer 2 Report

The authors have made sufficient changes based on the previous comments. 

Author Response

The authors have made sufficient changes based on the previous comments.

We thank the Reviewer for their positive decision.

Please kindly note that based on a previous suggestion from the Reviewer, we would like to propose representative histopathology images of SCCOHT from this study (currently included as Fig. 1A and Supplementary figure 1).

Reviewer 3 Report

The authors have reworked their manuscript to highlight significant data and drawn sound conclusions. The authors should be commended - took a significant effort restructure and reorganize. However, the manuscript reads well and is a sound contribution to the scientific literature.

Author Response

We thank the Reviewer for these encouraging comments.

Round 2

Reviewer 1 Report

The authors made substantial changes and revised their manuscript. It can be published in the present form. However, they should check the provided version of the supplementary tables file. Data mentioned in their rebuttal letter are not included.

This manuscript is a resubmission of an earlier submission. The following is a list of the peer review reports and author responses from that submission.

Round 1

Reviewer 1 Report

Aurélie and colleagues performed a comprehensive genomic and functional investigation of SCCOHTs. The data and experiments here, particularly the functional studies and targeted therapeutic results, are interesting and contribute to the field. Unfortunately, the WES analysis is flawed as many of the “mutations” discovered in SCCOHT are well-described benign polymorphisms reported in genomic databases (e.g. dbSNP, gnomAD, ExAC). This confounds the analysis, and as is currently presented, casts doubt on the significance of the WES results and conclusions.

1. The authors need to provide complete details when reporting genomic variants (PMID: 26931183). This can be added as a supplemental table. These data should be readily available in the exome variant call files.

2. ALL of the 14 variants described in the original cohort of 6 samples are present on chromosome 19p and many of the variants for which the results and discussion have been centered are benign polymorphisms (examples below). In fact, 4 of the 5 PLK5 variants listed in supplemental Table S5 are benign variants.

ABC7 p.R1349Q (https://www.ncbi.nlm.nih.gov/SNP/snp_ref.cgi?type=rs&rs=rs3745842)

PLK5 p.P320del (https://www.ncbi.nlm.nih.gov/SNP/snp_ref.cgi?type=rs&rs=rs58035688)

The high VAFs and the passing of the filter “mutated allele frequency higher in tumor than in normal tissue” is a result of the LOH or CN-LOH at 19p. The authors will need to redo this analysis to exclude common SNPs. This greatly impacts their results, interpretation, and conclusions.

3.  Figure 1: Do the NCI:H1299 cancer cell lines harbor 19p LOH? This again may explain the high VAF of the benign R314Q polymorphism in the cell line. Another interpretation of these data are that cells with 19p LOH are sensitive to PLK1 inhibitors. Again, it is unlikely that the variant PLK5 p.R314Q variant itself is deleterious since 11% of the population is homozygous, and 44% of the population is heterozygous. A control using a SMARC4-mutated/19p diploid cell line is needed.

4. Lines 320-321, the authors should cite previous reports of 19p CN-LOH in SCCOHT (PMID: 30275002)

Reviewer 2 Report

The goal of this study is to perform integrated genome-wide analysis for SCCOHT which is highly significant. However, the manuscript has many editorial mistakes and some key experiments are missing. The quality of many figures are low and are not well described. The order of the figures is confusing and hard to follow.

1.      qRT-PCR and/or western should be performed to validate RNA-Seq and SCNAs data (Tables 1C and 3A), especially the ones that have a high Polyphen2 score.  

2.      Typos or mistakes:

Line 135: S2A; not 2A

Line 153: Table; not Tables

Line 170: Table 1B and 1C; not Fig

Line 246: Should it be IGR-03, not -04?

Line 268: Table S6; not Table 6

Line 274: Fig 3C; not 5C

Supplement material and methods page 2 last row: Table S7; not table S8

3.      Table 1C needs a better description of all the color codes and symbols used. Do different colors in the gene structures represent different domains? If so, what is the significance of each domain? What does “na” (or “NA”; not consistent in the table) mean in Polyphen2 score and RNA-Seq? Were they not tested? No results? Not applicable (why?)? What do “-“ and “*” mean? Why don’t some frame shift mutations have a Polyphen2 Score (should be high presumably).  

4.      Line 161: there are only 6 mutations in Table 1C, but text said 7/8 tumors were detected by RNA-seq. Which mutation was found in multiple tumors?

5.      Lines 191 and 192: Authors should explain how 29% and 20% came out from the data in Table S5.

6.      Line 196: Authors should explain why they used PLK1 inhibitor instead of PLK5. Does this inhibitor also affect PLK5 non-specifically?

7.      Authors should transfect WT PLK5 in H1299 cells to see if it will rescue the anti-proliferation effect caused by the inhibitors.

8.      Table 2: Authors should provide the formula for calculating the mutations frequency because some of the numbers do not make sense.

9.      Figure 2 needs a better ligand. There is no label of what the X and Y axis are. The figure resolution is poor.

10.   Lane 241: Is IGR-01 the one that demonstrated a modest gain in 19p? It should be listed in the text.

11.   Table 3: Authors should explain what the grey boxes mean (the start and the end of common LOH region?). The quality of Table 3 is really poor. What do those wavy underlines mean in Table 3B?

12.   Line 251/252: except for IGR-03, which authors did not mention nor provide an explanation.

13.   Line 256: Which 5 samples are SMARCA4-M+ and which 3 are WT? SMARCA2 was downregulated in 7 out of 8 tumors, so apparently SMARCA2 expression is not completely related to SMARCA4 status.

14.   Line 258: Why did authors use these less known house keeping genes and how did they “normalized” 3 ratios in real time RT-PCR? Need explanations.

15.   [In general] “RT-PCR” means traditional non-quantitative RT-PCR (run on gels). If the authors did real-time RT-PCR, they should state “qRT-PCR” or “real-time RT-PCR” in the whole text.

16.   Figure 3C need much better labeling and more description. Authors failed to describe/explain all the “+” and “-“ in the boxes. The labels of x- and y-axis, the shades of the bars, the concentration unit of the drug, the time unit of the incubation should be included. NIH-Ovcar3 data were not explained in the text.

17.   Supplementary Table S2A was not referred by any text.

18.   [In general] Authors use (Figure XX), (Fig. XX) and (Fig XX) to refer a figure in the text. It needs to be consistent.  

Reviewer 3 Report

The manuscript entitled "Small cell carcinoma of the ovary, hypercalcemic type (SCCOHT) beyond SMARCA4 mutations: a comprehensive genomic analysis" is a very good study.

The authors conducted whole exome sequencing, RNA-sequencing and array comparative
genomic hybridization in SCCOHT to identify other recurrent genomic alterations. The authors reported  genomic stability and low mutation load, importantly the genomic alterations are highly recurrent. Very interesting manuscript with novel set of data.

Comments:

 The ethical approval number to use human samples needs to be included under section 2a.

Discussion: The authors may discuss the available/proposed/under clinical trial medications (tazemetostat; ponatinib; alisertib etc) a little.

Results: Please include the histopathology of SCCOHT of the samples analysed.

The authors analysed SMARCA2/4, but did not confirmed the antigenic reactivity of the tumor cells that were negative for SMARCA4 expression. They need to study SMARCB1

Reviewer 4 Report

Aurélie et al. report in their manuscript about the characterization of genomic aberrations in Small cell carcinoma of the ovary, hypercalcemic type (SCCOHT). This tumor type is very rare but has a rather low survival rate thus representing a target for both basic and clinical research to improve the prognosis of affected patients. The knowledge about genomic aberrations besides SMARCA4 mutations is limited. Therefore the clinical need and the limited knowledge justify the presented work. The authors conducted whole exome sequencing, RNA sequencing and CGH analyses to identify aberrations. Validations were partially done by targeted analyses using other samples or functional studies with exemplary cell cultures. Despite interesting data this reviewer sees major limitations of the study specifically for the described newly identified recurrent mutations. Moreover the manuscript contains several discrepancies between text and figures and missing information within the figures. Following remarks should be addressed by the authors.

Major remarks

The authors identified genomic variants within the SCCOHT samples compared to normal blood DNA and describe them general as mutations implying that these were de-novo mutations in the tumor cells. This is not correct as only 4/64 mutations (Suppl. Tab S4) showed an absence in the normal cells. Moreover all identified variants are located at Chr.19p13.2/3 the identified CN-LOH. Thus most variants were identified because of the loss of the second allele. It is likely that most variants are polymorphisms present in the population. Exemplarily, FBN3 variants G1614S, L1904F, P1958H and PLK5 variants L253V, R314Q and G323R were searched by the reviewer in public databases and all represent known SNPs with high allele frequencies in the population (even for the homozygous genotype). To identify tumor specific somatic mutations an absence (or a very low frequency) of this variant in the normal DNA is commonly required. Applying this criterion the TCGA ovarian cancer analysis resulted in the identification of 4 FBN3 mutations (as cited in Suppl. Tab S4) that are all unknown variants not SNPs. The authors have to carefully reanalyze their data (Suppl. Tab. S3, S5) to identify tumor-specific mutations. Most affected genes (except the known SMARCA4) will unlikely be recurrently mutated (>50%) within the analyzed samples.

To analyze potential therapeutic consequences of PLK5 mutations the authors analyzed two SMARCA4-mutated cell lines with or without PLK5 R314Q variant (BIN-67 and NCI-H1299) for their sensitivity against PLK1 inhibitors. To evaluate these data the authors must at least include data for the PLK1 status of the analyzed cell lines to exclude any bias by different effects caused by the PLK1 influence. Additionally, to support their hypothesis that PLK5 variants have functional consequences and can be targeted knockdown/overexpression experiments would be required.

The authors should correct discrepancies through the manuscript. (i.e. Suppl. Tab S3: only 4 SMARCA4 mutations included; Line 311: 100% validation rate by RNA-Seq not correct). Furthermore genes not validated by RNA-Seq should be excluded from further experiments (i.e. Tab.2: 12 gene included, only 11 validated)

Minor remarks

Line 127: typing error or => for

Line 227: change “expressed” to “detected”

Line 244: “significant SCNAs” - was the significance tested? If not, change to “aberrant SCNAs”

Line 246: Why shows IGR04 not the common deletion? Maybe IGR03?

Figure 3C: Add the missing labelling of x-, y-axis and the left tables.

Line 263 and 265: “up-regulation in RHOA…downregulation in RB” to “upregulation of RHOA…downregulation of RB”

Please add a correct legend for Suppl. Tab S3